# Measles Sequencing: Lessons Learned from a Large-Scale Outbreak

**DOI:** 10.3390/v17070913

**Published:** 2025-06-27

**Authors:** Victoria Indenbaum, Efrat Bucris, Keren Friedman, Tatyana Kushnir, Hagar Eliyahu, Roberto Azar, Tal Levin, Yara Kanaaneh, Eric J. Haas, Shepherd Roee Singer, Yaniv Lustig, Ella Mendelson, Oran Erster, Neta S. Zuckerman

**Affiliations:** 1Central Virology Laboratory, Public Health Services, Ministry of Health, Chaim Sheba Medical Center, Ramat Gan 52621, Israel; efrat.bucris@sheba.health.gov.il (E.B.); oran.erster@sheba.health.gov.il (O.E.); 2Faculty of Health Sciences, Ben Gurion University of the Negev, Beer Sheva 8410501, Israel; 3Division of Epidemiology, Public Health Services, Ministry of Health, Jerusalem 9438317, Israel; 4Hebrew University-Hadassah Braun School of Public Health and Community Medicine, Jerusalem 9112001, Israel; 5School of Public Health, Tel-Aviv University, Tel-Aviv 69978, Israel

**Keywords:** measles virus, whole-genome sequencing, molecular epidemiology, outbreak, phylogenetics

## Abstract

Between 2018 and 2019, Israel experienced one of its largest measles outbreaks in recent decades, with over 4300 reported cases and more than 100 documented importation events. Despite high national vaccination coverage, the prolonged nature of the outbreak posed a risk to the country’s measles elimination status. Traditional epidemiological investigations and genotyping based on the N450 region lacked sufficient resolution to differentiate between sustained local transmission and multiple independent introductions. To address this, we performed whole-genome sequencing on 123 measles virus samples representing both imported and locally acquired cases from diverse geographic regions. Phylogenetic analysis revealed multiple, distinct transmission chains, several of which could be linked to separate importation events. The MF non-coding region (MF-NCR) not only showed the highest genetic variability, but also contained many of the phylogenetic cluster-defining mutations, though informative changes were found throughout the whole genome. These findings demonstrate the value of whole-genome sequencing in resolving complex transmission dynamics and highlight the importance of integrating genomic epidemiology into routine measles surveillance. Such integration can enhance outbreak investigations and better inform public health responses to protect elimination status.

## 1. Introduction

In recent years, measles has re-emerged as a global public health concern, despite widespread implementation of vaccination programs and decades of progress in disease control. Between 2018 and 2019, a sharp resurgence in measles incidence occurred worldwide, with the World Health Organization (WHO) reporting over 800,000 cases and an estimated 207,500 deaths in 2019 alone—the highest annual measles mortality reported in over two decades [1]. This resurgence represented a reversal of earlier trends and underscored the fragility of measles elimination efforts, even in high-income countries with established public health infrastructure. Contributing factors included stagnating or declining vaccination rates, disruptions in healthcare delivery systems and the growing influence of vaccine hesitancy in certain populations [2]. The COVID-19 pandemic further exacerbated these challenges, disrupting routine immunization services and amplifying global susceptibility to measles [3,4,5].

Between 2017 and 2019, Europe experienced repeated waves of measles outbreaks and transmissions, ultimately leading to the loss of measles elimination status in several countries in 2019, including Albania, the Czech Republic, Greece, and the United Kingdom [6]. This loss was attributed to the re-establishment of endemic measles transmission, defined as continuous virus circulation for 12 months or more. During this period, molecular surveillance identified genotype D8 as the predominant circulating measles virus (MeV) in the WHO European region, with most cases associated with the named strain MVs/Gir Somnath.IND/42.16 [7].

Israel was similarly affected during this period. From March 2018 to July 2019, the country experienced one of its largest measles outbreaks in decades, resulting in over 4300 confirmed cases and widespread transmission across multiple districts and demographic groups. Despite maintaining high national vaccination coverage, the outbreak exploited immunity gaps in specific subpopulations. Epidemiological investigations identified over 100 distinct importation events during this period, many involving travelers returning from regions experiencing endemic or epidemic measles circulation [8]. Nonetheless, despite numerous documented importation events from various international sources, molecular surveillance relying on standard Sanger sequencing of the N450 fragment revealed that most cases were linked to genotype D8, specifically the named strain MVs/Gir Somnath.IND/42.16. This pattern raised concerns about ongoing local transmission, potentially threatening the measles elimination status that Israel had maintained since 2014.

During the outbreak in Israel, the rapid surge in case numbers hindered traditional epidemiological efforts to trace transmission chains. Although N450 sequencing is the global standard for measles genotyping, it provided limited resolution, often grouping genetically indistinguishable cases into broad clusters without clear epidemiological connections. As a result, it was not possible to determine whether the outbreak was driven by multiple, distinct transmission events following numerous importations, or by a single, prolonged chain of local transmission posing a risk to elimination status.

Whole-genome sequencing (WGS) of MeV provides significantly higher resolution than N450-based genotyping and has emerged as a powerful tool for investigating MeV transmission dynamics. Studies in Canada and the United Kingdom have demonstrated the utility of WGS in identifying superspreading events, uncovering silent transmission chains, and distinguishing co-circulating viral lineages that were indistinguishable by N450 sequencing alone [9,10,11]. These advances enable more precise assessments of measles outbreak dynamics and can support more targeted and timely public health interventions, particularly in regions at risk of repeated importation events.

In this study, we retrospectively applied WGS to a subset of clinical samples collected across Israel during the 2018–2019 measles outbreak. Our aim was to achieve higher-resolution characterization of the outbreak dynamics and to better understand transmission patterns—particularly in relation to the numerous importation events and their potential role in seeding multiple independent transmission clusters, as opposed to a single, prolonged chain of local transmission.

## 2. Materials and Methods

### 2.1. Sample Collection and Selection

During the 2018–2019 measles outbreak, a total of 4311 cases were reported, including 118 classified as importations [8] (updated data from the Israeli Ministry of Health, internal communication). Of these, 2191 cases were laboratory-confirmed at the Israeli National Measles, Mumps, and Rubella Laboratory. For this study, 123 samples were selected for WGS. This included 11 samples that were identified as importations based on epidemiological investigations conducted by the Israel Ministry of Health’s Division of Epidemiology, and successfully processed for WGS, along with 112 additional samples randomly chosen from the remaining laboratory-confirmed cases. The selection was designed to ensure broad geographic representation across District Health Offices (DHOs).

For each sample included in the WGS analysis, the sampling date, DHO-associated city, and—when relevant—the country of importation were recorded. In cases where the original sampling date was unavailable, the laboratory receipt date was used as a proxy, as it typically closely followed sample collection.

### 2.2. MeV Whole Genome Sequencing

Total nucleic acids were extracted from clinical samples selected for whole-genome sequencing. Reverse transcription and amplification of the complete MeV genome were performed in a single one-step RT-PCR reaction using a published set of genotype D8-specific primers designed to amplify the entire viral genome [9]. To improve coverage of the MF-NCR region, we supplemented the primer set with an additional published set of primers specifically targeting this region [12]. To further improve coverage of the complete MeV genome, we designed and added five new oligomers to the primer mix, as detailed below:
Primer MeV-L19-1F, sequence: 5′-CTAACCGATCATATCAAGGCAGAG-3′Primer MeV-L19-1R, sequence: 5′-GTCCGCACAGATGATTCAATTATC-3′Primer MeV-L20-F1, sequence: 5′-CACGTGGGTAGGCAGTATAGATTG-3′Primer MeV-L20-R1, sequence: 5′-ACCTGACAAAGCTGGGAATAG-3′Primer MeV-L21-1F, sequence: 5′-CAAAGAAGTCAACAAGGGATGTTC-3′


PCR products were then used to prepare DNA libraries using the Nextera XT library preparation kit (Illumina, CA, USA), following the manufacturer’s protocol. Paired-end sequencing was conducted on the Illumina MiSeq platform (Illumina, CA, USA).

### 2.3. Bioinformatic Analyses

Raw fastq sequencing files underwent quality assessment using FastQC and MultiQC [13]. Low-quality reads were trimmed using Trimmomatic [14]. High-quality reads were then aligned to an annotated MeV genotype D8 reference genome (KT732261.1) using BWA-mem (v0.7.19) [15,16]. The resulting alignments were processed with the SAMtools suite for sorting, indexing, and additional quality control. Consensus genome sequences were generated using iVar (https://andersen-lab.github.io/ivar/html/index.html (accessed on 1 January 2025)), with positions supported by fewer than ten reads masked as ‘N’. Sequences with >80% coverage were considered in this study.

For the phylogenetic analysis, the Nextstrain Augur pipeline [17] was used. Sequences were aligned to the D8 reference genome using MAFFT [18], followed by construction of a time-resolved phylogenetic tree using IQ-TREE and TreeTime [19,20], employing the GTR substitution model. Final visualization was performed with Auspice.

To quantify genetic variation, aligned consensus sequences were analyzed in R, where the number of mutations per gene or genomic segment, based on the annotated KT732261.1 reference sequence, was calculated. The results were visualized in Microsoft Excel. Shannon entropy was computed via R (version 4.3.2) and visualized with Excel.

## 3. Results

### 3.1. A Prolonged Outbreak with Multiple Transmission Chains

To overcome the limitations of traditional tracing methods during Israel’s large and complex 2018–2019 measles outbreak, molecular epidemiology was employed to better resolve transmission dynamics. WGS was conducted on a temporally and geographically representative subset of measles cases (*n* = 123), including both locally acquired (*n* = 112) and imported (*n* = 11) infections (Table 1).

These data were used to reconstruct a phylogenetic tree (Figure 1), which illustrates both the geographic distribution of measles cases and the genetic relationships between them. The tree reveals multiple distinct transmission chains (Figure 1A–D). In some clusters, transmission appears to occur primarily within a single District Health Office (DHO), such as within Tel Aviv (Figure 1A). In contrast, other clusters demonstrate broader dissemination across multiple DHOs. For instance, Figure 1B shows the spread of a genetically similar MeV strain across six geographically distant DHOs, including the northern, Haifa, central, and Jerusalem districts. Additional examples include inter-district transmission between Ramla and Haifa (Figure 1C), and between Jerusalem and Tel Aviv (Figure 1D), demonstrating the spread of the virus across administrative and geographic boundaries.

### 3.2. Imported Cases Initiating Transmission Clusters

To further investigate the contribution of imported cases to local transmission, the phylogenetic tree was annotated with the country of importation for samples of patients returning from abroad (Figure 2). The analysis reveals multiple distinct transmission clusters initiated by discrete importation events at various points throughout the course of the outbreak, some of which led to limited spread while others resulted in broader chains of secondary transmission within Israel. In several examples illustrated below, imported sequences appear at the base of well-supported clusters, indicating their role in initiating local transmission (Figure 2A–D). For instance, a case reported in late July 2018 under the Zfat DHO in northern Israel, linked to importation from Ukraine, had an identical viral sequence to a local case reported in early August 2018 within the same DHO. This cluster also included additional local cases from geographically distant regions in Israel, with no known link to importation (Figure 2A). Another example is a case in Petach Tikva, central Israel, reported in October 2018 and linked to an importation from Ukraine; this case clustered with others from the central region and with a sample from the northern region reported later that year (Figure 2B). Similarly, a case associated with travel to the USA in April 2019 in the Tel Aviv region clustered with local cases from the same DHO reported around the same time (Figure 2C). Lastly, a case linked to an importation from Thailand in November 2018 clustered with a local case reported at a later time (Figure 2D).

### 3.3. WGS Improves Resolution of Measles Transmission

To assess the enhanced resolution offered by WGS in detecting mutations, we measured sequence variability at each position across the MeV genome among all analyzed samples. As anticipated, the MF-NCR region showed the highest variability, followed by other non-coding regions, while the H gene exhibited the lowest (Figure 3). To further explore how this variability contributes to phylogenetic structure, we annotated each branch of the phylogenetic tree with the gene or genomic region in which its defining mutations occurred (Figure 4). Notably, the MF-NCR not only displayed the greatest sequence variability, but also accounted for a large proportion of the mutations that defined major phylogenetic clusters. Even so, informative, cluster-defining mutations were observed across nearly all coding and non-coding regions of the genome, demonstrating the value of WGS in resolving transmission dynamics.

## 4. Discussion

The 2018–2019 measles outbreak in Israel lasted over a year, involved thousands of cases, and was associated with repeated importations—placing the country at risk of losing its measles elimination status. It was hypothesized that the outbreak did not result from a single, continuous chain of transmission, but rather from multiple, distinct transmission events, each likely seeded by separate importations. However, traditional epidemiological methods and N450 sequencing lacked the resolution needed to adequately address this hypothesis. This underscored the importance of applying WGS to achieve finer resolution, enabling more accurate differentiation between transmission chains and a clearer understanding of the outbreak’s dynamics.

The retrospective phylogenetic analysis of a selected subset of samples confirmed that the outbreak comprised multiple simultaneous transmission chains. Despite the limited number of imported cases included, the data revealed several clusters likely initiated by separate importation events, occurring at different and distant time points during the outbreak. These findings support the initial hypothesis that the outbreak did not stem from one prolonged chain of transmission, but rather from distinct, parallel introductions. Since the analysis was performed retrospectively and included only a limited number of the available imported cases, it was not possible to fully reconstruct all transmission chains or definitively associate each cluster with a specific importation event. The absence of data from many of the imported cases during the outbreak may have affected the phylogenetic clustering, leading to incomplete representation of the true origins and distinctness of certain transmission chains. In some cases, what appears to be a single transmission cluster may in fact represent several separate introductions of genetically similar viruses. For example, in Figure 2A, we show a cluster linked to an importation from Ukraine on 25 July 2018. However, a genetically similar case from 28 May 2018, prior to that importation, suggests that it may have resulted from an earlier, unsequenced importation from the same outbreak in Ukraine. Another example appears in Figure 2C, where cases from March 2019 are clustered with an importation from the USA dated 8 April 2018. It is likely that the March cases were actually linked to a different importation from the same outbreak in the USA that was not included in our analysis. This highlights the importance of sequencing as many imported cases as possible during an outbreak to more accurately resolve transmission chains.

To further explore how WGS contributed to enhanced resolution in the analysis of the 2018–2019 measles outbreak, the variability in each gene or non-coding region was assessed. Consistent with previous findings [9,21,22,23] the MF-NCR emerged as the most variable region across the sequenced genomes, followed by additional non-coding regions (Figure 3). Notably, however, many of the mutations which defined major transmission clusters within the phylogenetic analysis were observed within the MF-NCR (Figure 4), emphasizing its significant role in providing epidemiological information during outbreak investigations, when full-genome sequencing is not feasible. Nevertheless, additional informative mutations were also identified in various coding and non-coding regions across the MeV genome at key branching points in the phylogenetic tree. This supports the value of expanding the sequencing window as much as possible to achieve improved resolution between clusters and to gain deeper insight into transmission dynamics during outbreaks. Indeed, recent studies [24,25,26] further highlight the power of whole-genome sequencing in clarifying measles outbreak dynamics and elucidating transmission clusters. These findings underscore the importance of genomic epidemiology as a complementary tool to traditional epidemiological investigations, especially in contexts where contact tracing is incomplete or case information is scarce.

In conclusion, our findings highlight the advantages of integrating genomic epidemiology into routine measles outbreak investigations. By combining measles sequencing with epidemiological data, health authorities can more accurately reconstruct transmission patterns, differentiate between local spread and new importations, and better assess the risk to elimination status. As measles importations become more frequent due to global mobility, the routine application of genomic tools will be vital for timely, informed responses and sustained measles control.

## Figures and Tables

**Figure 1 viruses-17-00913-f001:**
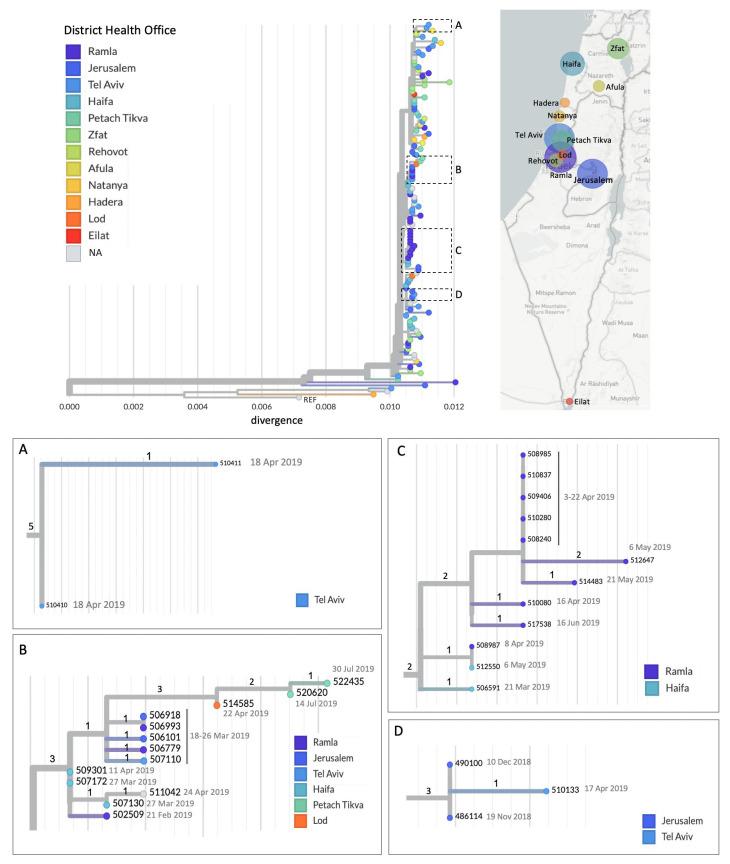
Phylogenetic tree highlighting transmission chains across geographic locations. Phylogenetic tree of 123 whole-genome MeV sequences from the 2018–2019 outbreak in Israel. Nodes are colored by geographic location, representing the district health office associated with each sequenced patient sample, as shown in the accompanying map. Circle sizes on the map reflect the number of sequences obtained from each location. The reference sequence (KT732261.1) is labeled as “REF”. Dashed boxes (**A**–**D**) and corresponding insets (**A**–**D**) highlight clusters representing distinct transmission chains occurring within single or multiple geographic locations. Sample collection date or—when unavailable—the date of receipt in the laboratory for MeV testing is indicated within each inset. Divergence scales vary across insets; the number of mutations is indicated by numbers along the branches.

**Figure 2 viruses-17-00913-f002:**
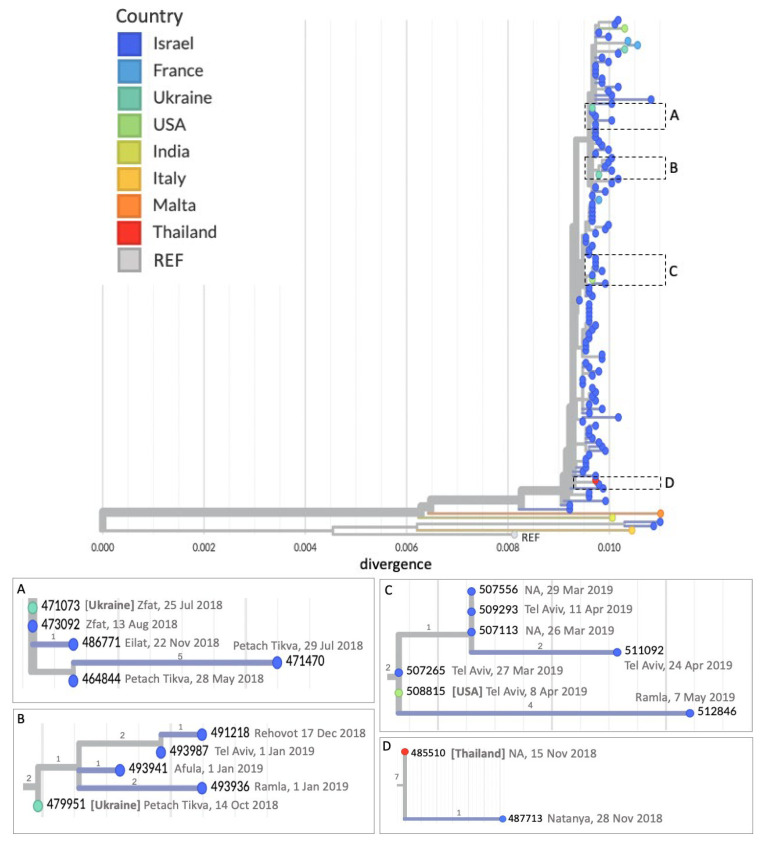
Phylogenetic tree highlighting importation-linked transmission. The phylogenetic tree in Figure 1, annotated according to country of importation linked to each sequenced patient sample or a local transmission. The reference sequence (KT732261.1) is labeled as “REF”. Dashed boxes (**A**–**D**) and corresponding insets (**A**–**D**) highlight clusters representing distinct transmission chains originating from sequenced patient samples linked to an importation. Sample collection date or—when unavailable—the date of receipt in the laboratory for MeV testing, as well as the DHO of the patient, is indicated within each inset. Divergence scales vary across insets; the number of mutations is indicated by numbers along the branches.

**Figure 3 viruses-17-00913-f003:**
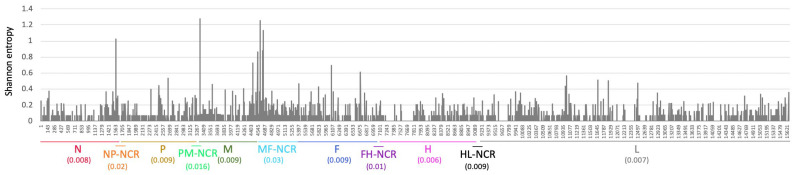
Genome-Wide Nucleotide Variability. Shannon entropy plot illustrating nucleotide variability at each position along the MeV genome, based on all sequenced data. Genomic regions are annotated along the x-axis, with the average entropy for each gene or non-coding region indicated in parentheses.

**Figure 4 viruses-17-00913-f004:**
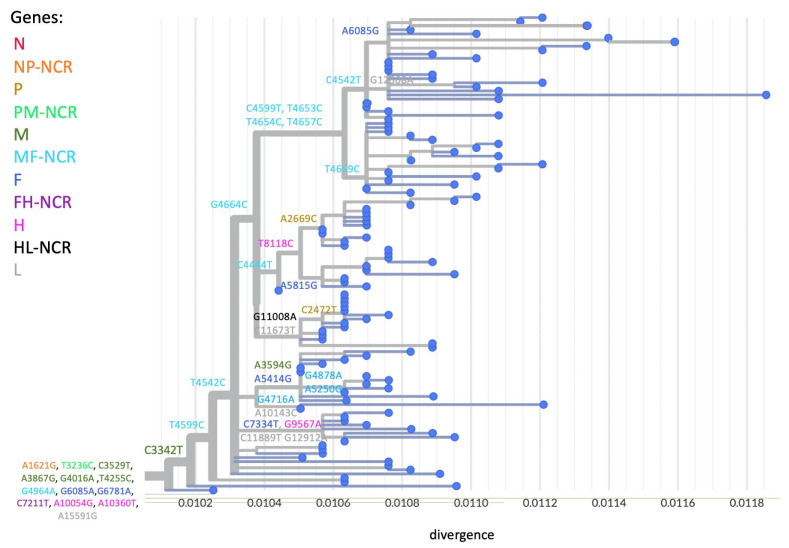
Genomic distribution of informative mutations in MeV phylogeny. The main branch of the phylogenetic trees shown in Figure 1 and Figure 2, annotated to indicate the MeV gene or genomic region in which key cluster-defining mutations occurred. Each gene or region is color-coded, with the specific mutation and its genomic position indicated.

**Table 1 viruses-17-00913-t001:** MeV outbreak samples selected for sequencing. The number of measles cases reported by each district health office (DHO) in Israel is presented, along with the number of identified importation events and their countries of origin reported by that district. The total number of cases per district is the number of cases and the number of imported cases.

District	DHO	# Cases	# Imported Cases
Northern	Zfat	9	1 (Ukraine)
	Afula	2	1 (USA)
Haifa	Haifa	12	1 (France)
	Hadera	1	1 (Italy)
	Natanya	2	1 (France)
Central	Lod	2	0
	Petach Tikva	9	1 (Ukraine)
	Ramla	21	1 (Malta)
	Rehovot	4	0
Tel Aviv	Tel Aviv	17	3 (France, USA, Ukraine)
Jerusalem	Jerusalem	20	0
Southern	Eilat	1	0
unknown		12	1 (Thailand)

## Data Availability

Sequences were submitted to GenBank.

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
