# Peer review of "Measles Sequencing: Lessons Learned from a Large-Scale Outbreak"

_viruses, 2025, doi:10.3390/v17070913_

Round 1
Reviewer 1 Report
Comments and Suggestions for Authors
A very well done study in molecular epidemiology and very relevant for Israel as well as many other countries in the immediate pre- and post measles elimination settings. The current method of genotyping of measles viruses may be adequate for countries in endemic settings, but lack the granularity needed to inform program decisions for countries closer to elimination.
However, a couple of comments.
(1) For figures 1 and 2 inset boxes A to D: It seems that the authors have used different horizontal scales for the inset boxes for better visualization of the data. This is fine but the authors should mention this specifically in the text below the diagram to prevent any miscommunication to the reader about genetic mutation distances in the phylogenetic tree.
(2) The different geographic zones in Israel/Jerusalem etc. affected by this continued outbreak form a complex geo-political territory. For example, for some clusters (Fig 2, inset box A), the more mutated 464844 virus in Petach Tikva appeared earlier than those in Zfat (471073 and 473092) and then there is a much longer chain appearing again in Petach Tikva (471470) within 4 days of the 471073 Ukraine virus in Zfat. Thus phylogenetic distances are not always congruent with time-space clusters. This can happen but the authors can consider highlighting the following points in the discussion.
(a) Measles being a purely person-to-person transmitted virus with no known extra human reservoir, mutation of the virus and onward transmission occurs through movement and interaction between sub-populations of humans, rather than merely through geographic contiguity. If there are more or less stringent restrictions (less transmission vs. more transmission) between human population movement between the complex sub-geographies in Israel, then the authors could consider mentioning this angle in the discussion section and thus enrich the data interpretation further.
(b) Usually, the principle of parsimony is followed in building a phylogenetic tree. In principle this is fine when the database (of the specimens analysed) is comprehensive. But if there are significant missing links, then the application of the parsimony principle should carry the caveat that the phylogenetic tree (especially the longer chains) could be different if more data were available. This can be mentioned in the discussion section.
Overall, an excellent paper and I recommend it for publication with these minor revisions.
Reviewer 2 Report
Comments and Suggestions for Authors
The authors have retrospectively applied whole genome sequencing ti a measles outbreak that occurred in Israel between 2018 and 2019. The identified several separate chains of transmission, most likely related to importation events from different geographic regions. The manuscript is well-written, with clear figures (although some of the colors were challenging, see below). The list of comments below is intended as suggestions for further improvement.
- Line 41: modify sentence to remove either “reporting” or “reported”.
- Lines 49/50: remove “, particularly …groups”.
- Lines 63/64: remove “- particularly …groups”.
- Lines 72-72: complicated and long sentence. Try to break up into two sentences.
- Line 90: remove “’s”.
- Line 100: briefly explain how these 11 samples were identified as (probable?) importation cases. Clinical measles within a week after arrival in Israel?
- Line 133: how many positions with coverage below 10 (masked as “N”) were accepted to still consider a sequence a “whole genome”?
- Figure 1: some of the colors are difficult to discriminate. Eilat looks red in the legend, but orange in the plot. Consider using a color scheme that provides more contrast, e.g. for color-blind people.
- Figure 2: in this figure the colors are even more challenging, as for obvious reasons the majority of the symbols are blue. It would be useful to also highlight the symbols above dashed box A, where it seems that importation was not associated with onward transmission in Israel. Or is this simply a limitation of the sample set selected (or available) for WGS?
- Figures 3 and 4: some of the light colors are hard to see on a white background.
- Line 228: change “characterized by” to “associated with”.
- Lines 229-231: in my opinion the authors should have chosen the inverse hypothesis, namely that there was endemic transmission of MeV over a period of one year. The authors should discuss whether or not the sample selection available for NGS was adequate for testing that alternative hypothesis.
- Line 290: GenBank accession numbers still to be added.
- The following additional references can be considered for the discussion: PMID 26153410, 29424220, 33662587, 39066448
- References 1, 6 and 7: I was unable to open the provided links. Check the links, or consider replacing with a reference to a publication in a peer-reviewed scientific journal.
- It is unclear to me what the author instructions of Viruses dictate, but I would recommend including the full list of authors (instead of first author followed by “et al.”) for an online journal.
